# Impact of matching error on linked mortality outcome in a data linkage of secondary mental health data with Hospital Episode Statistics (HES) and mortality records in South East London: a cross-sectional study

Amelia Jewell ,[1] Matthew Broadbent,[1] Richard D Hayes,[2] Ruth Gilbert,[3] Robert Stewart,[1,2] Johnny Downs [1,2]

For numbered affiliations see end of article.

**Correspondence to**
Amelia Jewell;
amelia.jewell@slam.nhs.uk

## ABSTRACT

**Objectives** Linkage of electronic health records (EHRs) to Hospital Episode Statistics (HES)-Office for National Statistics (ONS) mortality data has provided compelling evidence for lower life expectancy in people with severe mental illness. However, linkage error may underestimate these estimates. Using a clinical sample (n=265 300) of individuals accessing mental health services, we examined potential biases introduced through missed matching and examined the impact on the association between clinical disorders and mortality.

**Setting** The South London and Maudsley NHS Foundation Trust (SLaM) is a secondary mental healthcare provider in London. A deidentified version of SLaM's EHR was available via the Clinical Record Interactive Search system linked to HES-ONS mortality records.

**Participants** Records from SLaM for patients active between January 2006 and December 2016.

**Outcome measures** Two sources of death data were available for SLaM participants: accurate and contemporaneous date of death via local batch tracing (gold standard) and date of death via linked HES-ONS mortality data. The effect of linkage error on mortality estimates was evaluated by comparing sociodemographic and clinical risk factor analyses using gold standard death data against HES-ONS mortality records.

**Results** Of the total sample, 93.74% were successfully matched to HES-ONS records. We found a number of statistically significant administrative, sociodemographic and clinical differences between matched and unmatched records. Of note, schizophrenia diagnosis showed a significant association with higher mortality using gold standard data (OR 1.08; 95% CI 1.01 to 1.15; p=0.02) but not in HES-ONS data (OR 1.05; 95% CI 0.98 to 1.13; p=0.16). Otherwise, little change was found in the strength of associated risk factors and mortality after accounting for missed matching bias.

**Conclusions** Despite significant clinical and sociodemographic differences between matched and unmatched records, changes in mortality estimates were minimal. However, researchers and policy analysts

**Strengths and limitations of this study**

► The findings of our study demonstrate that although there are significant differences between matched and unmatched records, these did not significantly change the relative differences in mortality according to type of psychiatric disorder.

► The findings from this study are novel, in that data linkage error has not previously been evaluated in a routine mental health database, and this may be useful for other researchers looking to conduct research using electronic health records linked to Hospital Episode Statistics (HES)-Office for National Statistics (ONS) mortality data by National Health Service (NHS) Digital.

► A significant strength of the study is our access to the gold standard death data within the mental health record, which can be used to compare with the linked mortality data; this is useful as the mental health discrepancy in mortality is a particular focus on issues of parity/ inequality for health policy.

► Results may be generalisable to other NHS cohorts being linked to HES-ONS mortality data via NHS Digital as national standard matching methodology was used for the data linkage, and the quality and type of administrative data available for matching is likely to be similar within most NHS Trusts.

► False matches were not examined; therefore, total data linkage error within the linked dataset may be higher.

using HES-ONS linked resources should be aware that administrative linkage processes can introduce error.

## INTRODUCTION

Individuals with mental health disorders have substantially lower life expectancy in comparison with those without mental health disorders.[1 2] The UK government has been

working to reduce the mortality gap between those with and without mental health disorders by implementing policies aimed at improving the physical health of individuals with mental illness.[3] To inform changes in health policy, it is essential that we have good data available on health (physical and mental) and mortality in order to understand the trends and underlying mechanisms.

Electronic health record (EHR) data from the South London and Maudsley NHS Foundation Trust (SLaM) Clinical Record Interactive Search (CRIS) system[4] linked to Hospital Episode Statistics (HES) and death certificate data from the Office for National Statistics (ONS) have previously been used to provide evidence of the mental health mortality gap and investigate potential underlying mechanisms.[5–9] Data linkages such as that between CRIS and HES-ONS mortality resources are an important tool for increasing the use of existing data resources to support research.[10] However, the success of any data linkage and the quality of the subsequent linked dataset rely on multiple factors, including the data quality and linkage methodology. The two main types of linkage error that can occur during the data linkage process are false matches (ie, false positives) and missed matches (ie, false negatives). Errors in data linkage may result in systematic bias in reported outcomes.[11] False positives in data linkages can dilute the association between variables,[12] whereas false negatives can cause an underestimation of the risk outcome[13]; for example, a US study found that linking on an infant's medical insurance record, as opposed to the mother's, led to a significant underestimation in infant mortality due to the number of missed matches.[14]

There are a number of approaches to evaluating linkage quality, for example, by comparing characteristics of linked and unlinked records to identify potential sources of bias.[12] Previous studies have identified significant differences in sociodemographic characteristics between matched and unmatched samples in data linkages, including sex,[15–21] age,[15 19 22–27] ethnicity,[16 19–22 26–32] socioeconomic status,[20 27 33 34] marital status[15 22 28 35] and level of education,[20 34 35] providing some indication that linked data may not always be representative of the population of interest.[12] Reference or 'gold standard' datasets in which the true match status is known can be used to evaluate the impact of data linkage error by quantifying missed or false matches[36]; however, gold standard data either do not exist or are rarely available to researchers using linked data due to issues around governance and data protection.[12]

If linked data are being used to inform health policy, it is important that the effect of linkage error is estimated. At present, for example, it is not clear whether any potential error in the linkage between CRIS and HES-ONS mortality data has a bearing on the research being conducted using the linked data. It was therefore our aim to investigate the effect of linkage error on the prediction of a linked outcome measure in this mental health population by, first, examining the administrative, sociodemographic and clinical characteristics of matched and unmatched records, and second, by comparing the prediction of linked mortality outcome with gold standard death data provided by the local mental health record.

## METHODS

### The Mental Health Dataset – CRIS

The Maudsley Biomedical Research Centre (BRC) CRIS system, described previously in detail,[4 37] is a deidentified database of electronic medical records for mental health service users within SLaM: one of the largest providers of secondary mental healthcare in Europe. SLaM provides local area mental health services predominately for the London boroughs of Lambeth, Southwark, Croydon and Lewisham (the SLaM 'catchment area'), as well as some national specialist services.[4] Records from CRIS have previously been linked to a number of external databases including the National Pupil Database,[27 38] the Thames Cancer Registry[39] and general practice (GP) data.[40]

All individuals (children and adults) who were active in SLaM between 1 January 2006 and 20 December 2016, not including Improving Access to Psychological Therapies (IAPT) services, were included in the current study. Exposure variables (table 1) were collected from structured fields within the CRIS system and included sociodemographic (eg, age, sex, ethnicity and mortality), clinical (eg, mental health diagnosis, referral status and level of service use) and administrative variables (ie, patient identifiers available for linkage).

Mortality data from CRIS provided the 'gold standard' death data using both formal and informal notification sources.[41] All-cause mortality data are entered into the SLaM EHR either by the care team directly (eg, after being informed by family members) or via automatic tracing, whereby the SLaM Trust run a demographic batch trace weekly for all patients (past or current) against the NHS Summary Care Record (SCR). This returns date of death for those patients who have died (figure 1). The SCR is an electronic record of important patient information created from GP medical records that can be seen and used by any authorised staff in areas of the health system who are involved in a patients' direct care. The SCR is administered by NHS Digital.[42] Date of death in the SCR is entered at the point of occurrence, that is, the death may not have been formally registered yet.

### The HES and ONS mortality dataset

HES data are a national dataset, governed by NHS Digital, which holds details of all admissions, outpatient appointments, and accident and emergency attendances at NHS hospitals in England.[43] HES data are derived from the Secondary Uses Service database, which collects submissions routinely from all NHS acute hospital and mental health trusts in England. The data accuracy and quality within these submissions are important, especially to NHS Trusts, as they are used to calculate payments for the care

**Table 1** Exposure variables

| Exposure variable | Description |
| --- | --- |
| NHS number present | Was NHS number available for linkage? Binary variable (yes vs no). |
| Date of birth present | Was date of birth available for linkage? Binary variable (yes vs no). |
| Sex present | Was sex available for linkage? Binary variable (yes vs no). |
| Postcode present | Was postcode (full or part) available for linkage? Binary variable (yes vs no). |
| Age | Age in years at time of PII extraction (20th December 2016). |
| Sex | Sex, male versus female. |
| Patient deceased | Is the patient deceased according to the gold standard CRIS-derived mortality data (yes vs no)? |
| Ethnicity | Patient ethnicity, coded into six categories: (1) British, Irish or any other white ethnic groups, (2) mixed, (3) Indian, Pakistani, Bangladeshi or 'other Asian', (4) Caribbean, African or 'other black', (5) other and (6) not stated. |
| Resident in SLaM catchment area | Was the patient resident in the SLaM catchment area, that is, Lambeth, Lewisham, Southwark or Croydon, at the time of PII extraction, binary variable (yes vs no)? |
| Quartiles of neighbourhood deprivation | Quartiles of deprivation score: first (most deprived), second, third and fourth (least deprived). |
| Referral status in past 2 years | Referral status in the 2 years prior to PII extraction: (1) accepted, (2) discharged, (3) rejected or (4) no referral in 2 years prior to linkage. |
| Primary diagnosis ever | A primary diagnosis ever of F00-F09: organic, including symptomatic, mental disorders, F10-F19: mental and behavioural disorders due to psychoactive substance use, F20-F29: schizophrenia, schizotypal and delusional disorders, F30-F39: mood (affective) disorders, F40-F49: neurotic, stress-related and somatoform disorders, F50-F59: behavioural syndromes associated with physiological disturbances and physical factors, F60-F69: disorders of adult personality and behaviour, F70-F79: mental retardation, F80-89: disorders of psychological development, F90-F98: behavioural and emotional disorders with onset usually occurring in childhood and adolescence, or other diagnosis. Binary variables (yes vs no). Other diagnosis category includes all non-mental health (ie, non F code) ICD-10 diagnoses as well as 'F99: Mental disorder, not otherwise specified' and 'F00-F99: Mental and behavioural disorders'. Patients can have multiple primary diagnoses throughout their time at SLaM and therefore can appear in multiple groups. |
| Face-to-face contact | Quartiles of face-to-face contact with SLaM from 1 January 2006 to 20 December 2016; first (least face to face contact), second, third and fourth (most face to face contact). |
| Inpatient bed days | Number of SLaM inpatient bed days from 1 January 2006 to 20 December 2016, coded into: (1) none (0), (2) low (1–2 days), (3) moderate (3–31 days) and (4) high (32+ days). |
| Optimal match | Binary variable (optimal match vs non-optimal match). For those records that were successfully matched by NHS Digital only. An optimal match represents a 'perfect' match or a match rank of one (see table 2), that is, the records matched on all supplied patient identifiers (NHS number, sex, date of birth and postcode). |

CRIS, Clinical Record Interactive Search; ICD-10, International Classification of Diseases, 10th revision; NHS, National Health Service; PII, Patient identifiable information; SLaM, South London and Maudsley NHS Foundation Trust.

provided.[44] HES data are designed to enable secondary use, for example, for research, and can be linked to other external data sources.

The ONS mortality data consists of the Primary Care Mortality Database (PCMD), which includes date and cause of death for all deaths registered in England and Wales. The data are collected by the ONS, and access to the data is managed by NHS Digital.[45] Mortality data from the PCMD are only returned for individuals who have matched and have died; individuals without a mortality record are therefore assumed to be alive.

## Linkage procedures

The linkage between CRIS and HES-ONS mortality data under analysis here was conducted in January 2017.

Patient identifiable information (PII), that is, date of birth, sex, NHS number and postcode, for all SLaM records (excluding anybody who had previously opted out of CRIS) was extracted on 20 December 2016. The data linkage was conducted by NHS Digital following deterministic matching procedures. Date of birth, sex, NHS number and most recently recorded postcode were used as the personal identifiers to match records. Table 2 demonstrates the match ranks used by NHS Digital, with the quality of the matching decreasing from step one to eight.

Initially records are matched to the Personal Demographics Service (PDS) using this method. The PDS is the national electronic database of NHS patient details

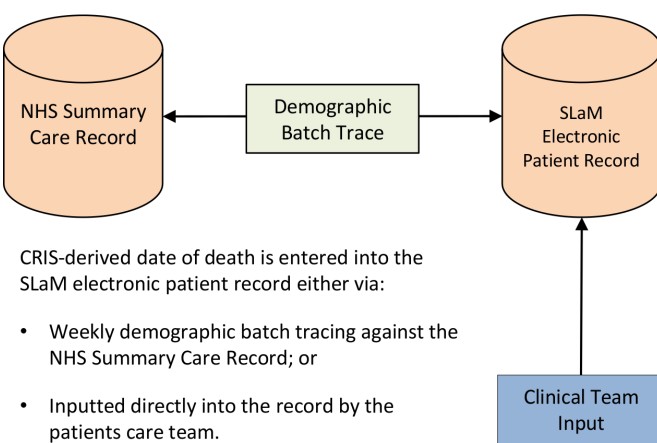

CRIS-derived date of death is entered into the SLaM electronic patient record either via:

- Weekly demographic batch tracing against the NHS Summary Care Record; or

- Inputted directly into the record by the patients care team.

**Figure 1** Flow diagram describing how the gold standard CRIS-derived date of death is entered into the SLAM electronic medical record. CRIS, Clinical Record Interactive Search; NHS, National Health Service; SLaM, South London and Maudsley NHS Foundation Trust.

such as name, address, date of birth and NHS number.[46] The PDS is also involved in the creation and updating of the NHS SCR, as described above. Theoretically, every CRIS record should match to the PDS, even if they do not have any HES episodes or a mortality record. Following matching to the PDS, patients who have opted out of their data being used for secondary purposes, known as 'type 2' opt-outs,[47] are removed from the cohort. A report is produced containing details of all the matched records, excluding unmatched records and opt-outs, known as the Flagging Current Status report. Mortality and HES data are then extracted for matched records only (see figure 2).

Match status outcome measure (ie, matched vs missed match) was determined using the Flagging Current Status report following the linkage of CRIS records by NHS Digital. Unmatched records included both records that were not matched (ie, missed matches) as well as opt-outs.

For those records that were successfully matched, we examined optimal match rank, whereby an optimal match represents a 'perfect' match or a match rank of one (see table 2), that is, the records matched on all supplied patient identifiers. The optimal match variable was provided by NHS Digital within the HES data following completion of the data linkage.

### Patient and public involvement

Patient and public involvement was key to the initial development and ongoing oversight of the CRIS system. The data linkage between CRIS and the HES-ONS mortality data, as well as this project specifically, were approved by the service user chaired CRIS Oversight Committee (project reference: 17–065). The CRIS, HES-ONS mortality data linkage was also presented to the Data Linkage Service User and Carer Advisory Group,[48] a regular meeting of people with lived experience of mental illness, all of whom have an interest in mental health research involving data linkage. The group provide ongoing advice and feedback to researchers conducting projects using the linked CRIS and HES-ONS mortality data.

### Statistical analysis

#### Step 1: missed matches analysis

Data were analysed using STATA V.15. Univariable logistic regression was first performed on all administrative, sociodemographic and clinical characteristics to examine the association with match status outcome for HES-ONS linkage (ie, those with a positive match on the Flagging Current Status report). Multivariable logistic regression was then performed to identify factors which remained significant predictors of HES-ONS matching after controlling for all other examined variables.

#### Step 2: optimal match analysis

In order to determine which variables predicted an optimal match, univariable logistic regression analysis

**Table 2** NHS Digital deterministic matching steps

| Match rank | NHS number | Date of birth | Sex | Postcode* | |
|---|---|---|---|---|---|
| 1 | Exact | Exact | Exact | Exact | |
| 2 | Exact | Exact | Exact | | |
| 3 | Exact | Partial | Exact | Exact | |
| 4 | Exact | Partial | Exact | | |
| 5 | Exact | | | Exact | |
| 6 | | Exact | Exact | Exact | Where NHS number does not contradict the match and DOB is not 1 January and the postcode is in the 'ignore' list.† |
| 7 | | Exact | Exact | Exact | Where NHS number does not contradict the match and DOB is not 1 January. |
| 8 | Exact | | | | |

*Current or most recently recorded postcode.
†The 'ignore' list includes postcodes for communal establishments such as hospitals, care homes, prisons and boarding schools.
DOB, date of birth; NHS, National Health Service.

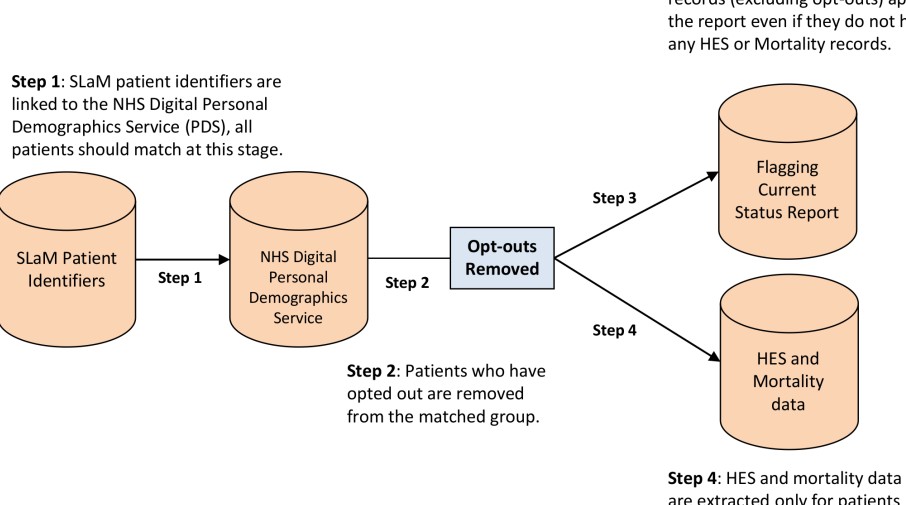

**Step 1**: SLaM patient identifiers are linked to the NHS Digital Personal Demographics Service (PDS), all patients should match at this stage.

**Step 3**: A flagging current status report is produced. All matched records (excluding opt-outs) appear in the report even if they do not have any HES or Mortality records.

**Step 2**: Patients who have opted out are removed from the matched group.

**Step 4**: HES and mortality data are extracted only for patients who matched in Step 1 and who have not opted out.

**Figure 2** Flow diagram of the linkage process between SLaM records and HES-ONS mortality data. HES-ONS, Hospital Episode Statistics-Office for National Statistics; NHS, National Health Service; SLaM, South London and Maudsley NHS Foundation Trust.

was performed on a subsample of the total cohort (ie, only those records which were matched with HES-ONS). Multivariable logistic regression was then performed to identify factors associated with an optimal match outcome after accounting for all other examined variables.

### Step 3: CRIS-derived mortality analysis

In order to examine the effect of potential matching bias on mortality outcome, we first examined factors associated with all-cause mortality using the gold standard CRIS-derived death data in univariable and multivariable analyses.

### Step 4: linked mortality analysis

We then examined predictors of mortality using the linked mortality outcome (ie, patients with a death date in the HES-ONS matched dataset), in the HES-ONS matched group only, in univariable and multivariable analyses. We used inverse probability weighting calculated from the step 1 analysis and the optimal match variable to adjust the final model for missed matching bias.

### RESULTS

Within the total cohort (n=265 300), 93.7% (n=248 698) of patients were successfully matched by NHS Digital to the HES-ONS data via the PDS. The mean age for the matched sample was 43.40 (range 5 months–117 years; SD 22.69), at the time the patient identifiers were extracted, with half the sample being male (50.0%). The majority of the matched group (58.5%) were of white ethnicity and were resident in the four London Boroughs serviced by SLaM, that is, Lambeth, Southwark, Lewisham and Croydon (73.4%). The 'Strengthening The Reporting

of Observational studies in Epidemiology' checklist[49] is reported in online supplementary material table 1.

### Step 1: missed matches analysis

Table 3 provides the administrative, sociodemographic and clinical characteristics of the CRIS sample, according to matched and unmatched status. With regards to the administrative variables examined (ie, NHS number, date of birth, sex and postcode present in PII sent to NHS Digital), records that did not have an NHS number or date of birth were not matched (n=5755, 2.17%). Records where sex was present in the PII sent to NHS Digital were eight times more likely to be matched than records where sex was not present (OR 8.14; 95% CI 4.66 to 14.23; p<0.001). Similarly, records where a postcode was present were almost seven times more likely to be matched in comparison with records where no postcode was available for matching (OR 6.68; 95% CI 6.26 to 7.12; p<0.001). Patients who had died prior to the PII being extracted, that is, 20 December 2016 (according to the gold standard CRIS-derived death data) were significantly more likely to match than patients who were alive.

Within the adjusted analysis, we found the likelihood of matching was significantly associated with a number of sociodemographic and clinical factors. Males and patients of non-white ethnicities, that is (1) mixed, (2) Indian, Pakistani, Bangladeshi or 'other Asian', (3) Caribbean, African or 'other black' or (4) other ethnicity, were all significantly less likely to match. Individuals without a stated ethnicity were also significantly less likely to match. Compared with patients in the lowest quartile of deprivation, patients in the second quartile were significantly less likely to match. We found no significant differences

**Table 3** Logistic regression analysis examining administrative, sociodemographic and clinical associations with match status (match vs missed match)

| Variable | Total population (n=265 300) | Matched (n=248 698, 93.74%) | Missed match (n=16 602, 6.26%) | OR (95% CI) | aOR† (95% CI) |
|---|---|---|---|---|---|
| **Administrative variables** | | | | | |
| NHS number present: n (%) | 259 545 (97.83) | 248 698 (100.00) | 10 847 (65.34) | | |
| Date of birth present: n (%) | 265 010 (99.89) | 248 698 (100.00) | 16 312 (98.25) | | |
| Sex present: n (%) | 265 246 (99.98) | 248 663 (99.99) | 16 583 (99.89) | 8.14 (4.66 to 14.23)** | |
| Postcode present: n (%) | 260 466 (98.18) | 245 278 (98.62) | 15 188 (91.48) | 6.68 (6.26 to 7.12)** | |
| **Sociodemographic variables** | | | | | |
| Age: mean (SD) | 43.40 (22.69) | 43.50 (22.79) | 41.92 (21.09) | 1.003 (1.002 to 1.004)** | 0.999 (0.998 to 1.000) |
| Male Sex: n (%) | 132 730 (50.04) | 123 818 (49.79) | 8912 (53.74) | 0.85 (0.83 to 0.88)** | 0.93 (0.89 to 0.96)** |
| Patient deceased: n (%) | 30 173 (11.37) | 29 057 (11.68) | 1116 (6.72) | 1.84 (1.73 to 1.95)** | 1.41 (1.30 to 1.53)** |
| Ethnicity: n (%) | | | | | |
| British, Irish or any other white ethnic groups | 138 495 (58.54) | 131 150 (58.93) | 7345 (52.35) | (reference) | (reference) |
| Mixed | 6853 (2.90) | 6477 (2.91) | 376 (2.68) | 0.96 (0.87 to 1.07) | 0.82 (0.73 to 0.91)** |
| Indian, Pakistani, Bangladeshi or 'other Asian' | 10 889 (4.60) | 10 155 (4.56) | 734 (5.23) | 0.77 (0.72 to 0.84)** | 0.78 (0.71 to 0.85)** |
| Caribbean, African or 'other black' | 40 725 (17.21) | 38 013 (17.08) | 2712 (19.33) | 0.78 (0.75 to 0.82)** | 0.70 (0.66 to 0.74)** |
| Other | 16 650 (7.04) | 15 393 (6.92) | 1257 (8.96) | 0.69 (0.64 to 0.73)** | 0.77 (0.72 to 0.83)** |
| Not stated | 22 961 (9.71) | 21 354 (9.60) | 1607 (11.45) | 0.74 (0.70 to 0.79)** | 0.89 (0.83 to 0.95)* |
| Resident in SLaM catchment area: n (%) | 187 773 (73.42) | 177 766 (73.53) | 10 007 (71.47) | 1.11 (1.07 to 1.15)** | 0.97 (0.93 to 1.02) |
| Quartiles of neighbourhood deprivation: n (%) | | | | | |
| First (most deprived) | 63 476 (25.03) | 60 220 (25.09) | 3256 (24.00) | (reference) | (reference) |
| Second | 63 452 (25.02) | 59 786 (24.91) | 3666 (27.02) | 0.88 (0.84 to 0.93)** | 0.90 (0.86 to 0.95)** |
| Third | 63 449 (25.02) | 60 052 (25.02) | 3397 (25.04) | 0.96 (0.91 to 1.00) | 1.01 (0.95 to 1.07) |
| Fourth (least deprived) | 63 221 (24.93) | 59 973 (24.99) | 3248 (23.94) | 1.00 (0.95 to 1.05) | 1.06 (1.00 to 1.12) |
| **Clinical variables** | | | | | |
| Referral status in 2 years prior to linkage: n (%) | | | | | |
| Accepted | 12 041 (4.56) | 11 423 (4.61) | 618 (3.78) | (reference) | (reference) |
| Discharged | 45 228 (17.11) | 42 394 (17.10) | 2834 (17.33) | 0.81 (0.74 to 0.89)** | 0.87 (0.78 to 0.97)* |
| Rejected | 9628 (3.64) | 8921 (3.60) | 707 (4.32) | 0.68 (0.61 to 0.76)** | 0.88 (0.76 to 1.01) |
| No referral in 2 years prior to linkage | 197 414 (74.69) | 185 221 (74.70) | 12 193 (74.57) | 0.82 (0.76 to 0.89)* | 0.87 (0.79 to 0.95)* |
| Primary diagnosis ever: n (%) | | | | | |
| F00–F09: organic, including symptomatic, mental disorders | 25 249 (9.52) | 24 248 (9.75) | 1001 (6.03) | 1.68 (1.58 to 1.80)** | 1.37 (1.26 to 1.49)** |

Continued

**Table 3** Continued

| Variable | Total population (n=265 300) | Matched (n=248 698, 93.74%) | Missed match (n=16 602, 6.26%) | OR (95% CI) | aOR† (95% CI) |
|---|---|---|---|---|---|
| F10-F19: mental and behavioural disorders due to psychoactive substance use | 27 755 (10.46) | 26 158 (10.52) | 1597 (9.62) | 1.10 (1.05 to 1.16)** | 1.12 (1.05 to 1.21)* |
| F20-F29: schizophrenia, schizotypal and delusional disorders | 18 516 (6.98) | 17 323 (6.97) | 1193 (7.19) | 0.97 (0.91 to 1.03) | 1.02 (0.93 to 1.11) |
| F30-F39: mood (affective) disorders | 42 578 (16.05) | 40 515 (16.29) | 2063 (12.43) | 1.37 (1.31 to 1.44)** | 1.28 (1.21 to 1.36)** |
| F40-F49: neurotic, stress-related and somatoform disorders | 33 248 (12.53) | 31 599 (12.71) | 1649 (9.93) | 1.32 (1.25 to 1.39)** | 1.27 (1.19 to 1.35)** |
| F50-F59: behavioural syndromes associated with physiological disturbances and physical factors | 8743 (3.30) | 8388 (3.37) | 355 (2.14) | 1.60 (1.43 to 1.78)** | 1.49 (1.31 to 1.69)** |
| F60-F69: disorders of adult personality and behaviour | 6945 (2.62) | 6585 (2.65) | 360 (2.17) | 1.23 (1.10 to 1.37)** | 0.95 (0.84 to 1.07) |
| F70-F79: mental retardation | 3018 (1.14) | 2898 (1.17) | 120 (0.72) | 1.62 (1.35 to 1.95)** | 1.55 (1.26 to 1.89)** |
| F80-F89: disorders of psychological development | 7242 (2.73) | 6991 (2.81) | 251 (1.51) | 1.88 (1.66 to 2.14)** | 1.86 (1.61 to 2.15)** |
| F90-F98: behavioural and emotional disorders with onset usually occurring in childhood and adolescence | 17 011 (6.41) | 16 363 (6.58) | 648 (3.90) | 1.74 (1.60 to 1.88)** | 1.58 (1.45 to 1.74)** |
| Other diagnosis | 119 638 (45.10) | 112 556 (45.26) | 7082 (42.66) | 1.11 (1.08 to 1.15)** | 1.23 (1.18 to 1.28)** |
| Quartiles of face-to-face contacts: n (%) | | | | | |
| First (least face-to-face contact) | 76 602 (28.87) | 70 822 (28.48) | 5780 (34.82) | (reference) | (reference) |
| Second | 64 914 (24.47) | 59 811 (24.02) | 5103 (30.74) | 0.96 (0.92 to 0.99)* | 0.94 (0.89 to 0.99)* |
| Third | 59 337 (22.37) | 56 116 (22.56) | 3221 (19.40) | 1.42 (1.36 to 1.49)** | 1.15 (1.08 to 1.22)** |
| Fourth (most face-to-face contact) | 64 447 (24.29) | 61 949 (24.91) | 2498 (15.05) | 2.02 (1.93 to 2.12)** | 1.42 (1.33 to 1.52)** |
| Inpatient bed days: n (%) | | | | | |
| None (0) | 238 573 (89.93) | 223 527 (89.88) | 15 046 (90.63) | (reference) | (reference) |
| Low (1–2 days) | 1446 (0.55) | 1342 (0.54) | 104 (0.63) | 0.87 (0.71 to 1.06) | 0.86 (0.67 to 1.09) |
| Moderate (3–31 days) | 9825 (3.70) | 9133 (3.67) | 692 (4.17) | 0.89 (0.82 to 0.96)* | 0.90 (0.81 to 1.00)* |
| High (32+ days) | 15 456 (5.83) | 14 696 (5.91) | 760 (4.58) | 1.30 (1.21 to 1.40)** | 1.14 (1.03 to 1.26)* |

Missing data: age (n=275); sex (n=55); ethnicity (n=28 727); resident in local catchment area (n=9537); quartiles of neighbourhood deprivation (n=11 702); referral status (n=989).
*P<0.05. **P<0.001.
†Adjusted for all other variables listed in the table.
aOR, adjusted OR; NHS, National Health Service; OR, Odds Ratio; SLaM, South London and Maudsley NHS Foundation Trust.

between patients in the third and fourth quartiles of deprivation (p>0.05).

With regards to clinical variables, patients who had been discharged or who had not had a referral in the 2 years prior to the PII being extracted were significantly less likely to match than patients who had an accepted referral in the past 2 years. Similarly, patients with the most face-to-face contact (third and fourth quartiles) were significantly more likely to match, while patients in the second quartile of face-to-face contact were significantly less likely to match. Patients with the highest number of inpatient bed days (32+) were significantly more likely to match than those with no inpatient bed days, whereas patients with a moderate number of inpatient bed days (3–31) were significantly less likely to match than those with no inpatient bed days.

In terms of diagnosis, patients who had ever received a primary diagnosis within the majority F00-98 ICD-10 diagnosis codes (see table 3) were all significantly more likely to match than patients who had never received an ICD-10 F diagnosis. However, primary diagnoses of schizophrenia, schizotypal and delusional disorders (F20-F29) and disorders of adult personality and behaviour (F60-F69) were not significantly associated with match status.

### Step 2: optimal match analysis

For records that were matched (n=248 698), we examined predictors of an optimal match. An optimal match represents a 'perfect match', that is, where the records were matched on all four patient identifiers provided to NHS Digital (NHS number, sex, date of birth and postcode). Within the adjusted model (see online supplementary material table 2), older age, male sex, an 'other' or 'not stated' ethnicity, living in the SLaM catchment area, deprivation, referral status in the 2 years prior to the linkage, low levels of face-to-face contact and a mental health diagnosis were all significantly associated with an optimal match. All-cause mortality and number of inpatient bed days were not significantly associated with optimal match outcome.

### Step 3: CRIS-derived mortality analysis

Table 4 displays clinical and sociodemographic factors associated with mortality using the gold standard CRIS-derived death data. Within the fully adjusted model (aOR†) older age, male sex, deprivation, a primary diagnosis ever of organic, including symptomatic, mental disorders (F00-F09), mental and behavioural disorders due to psychoactive substance use (F10-F19), schizophrenia, schizotypal and delusional disorders (F20-F29), or mental retardation (F70-F79), amount of face-to-face contact and inpatient bed days were all associated with a significant increased risk of mortality.

While non-white ethnicity and a primary diagnosis ever of mood (affective) disorders (F30-F39), neurotic, stress-related and somatoform disorders (F40-F49), behavioural syndromes associated with physiological disturbances and physical factors (F50-F59), behavioural and emotional

disorders with onset usually occurring in childhood and adolescence (F90-F98) or other diagnoses were all associated with a significant decreased risk of mortality.

### Step 4: linked mortality analysis

We examined the associations between mortality and sociodemographic and clinical factors using the linked HES-ONS mortality data for the matched group only (n=248 698). Within the adjusted model (see table 5), we found that older age, male sex and the second, third and fourth (ie, least deprived) quartiles of deprivation were significantly associated with mortality. With regards to clinical variables, a primary diagnosis of organic, including symptomatic, mental disorders (F00-F09), mental and behavioural disorders due to psychoactive substance use (F10-F19), or mental retardation (F70-F79) and a low, moderate or high number of inpatient bed days were all significantly associated with an increased risk of mortality. Similarly, patients with higher levels of face-to-face contact (second and third quartiles) were significantly more likely to have died when compared with patients with the least amount of face-to-face contact.

Non-white ethnicity and a primary diagnosis of mood (affective) disorders (F30-F39), neurotic, stress-related and somatoform disorders (F40-F49), behavioural syndromes associated with physiological disturbances and physical factors (F50-F59), disorders of psychological development (F80-F89) and behavioural and emotional disorders with onset usually occurring in childhood and adolescence (F90-F98) all negatively associated with mortality.

After further adjustments, using the optimal match variable and an inverse probability weighting, calculated during the missed match analysis in step 1, to account for potential linkage bias, we found these associations with mortality persisted (table 5).

We compared the output of the step 3 and step 4 analyses (table 4 – aOR† and table 5 – aOR‡), that is, predictors of CRIS-derived mortality and predictors of linked ONS mortality in the matched group (controlling for optimal matching and matching probability). The two final models were largely the same with slight differences in some of the ORs. However, there was a difference between the models within the diagnosis variables. In the model predicting CRIS-derived mortality, a primary diagnosis ever of schizophrenia, schizotypal and delusional disorders (F20-F29) was significantly associated with all-cause mortality (OR 1.08; 95% CI 1.01 to 1.15; p=0.02). Whereas in the model predicting linked mortality in the matched group, a primary diagnosis of F20-F29 was not significantly associated with mortality (OR 1.05; 95% CI 0.98 to 1.13; p>0.05), despite the ORs being virtually identical (1.08 for the CRIS-derived death model compared with 1.05 for the linked death model). The only other difference between the two models was regarding the F60-F69 diagnosis variable. The association between a primary diagnosis of a disorder of adult personality and behaviour (F60-F69) and mortality in the CRIS-derived

**Table 4** Logistic regression analysis examining sociodemographic and clinical associations with gold standard CRIS-derived mortality

| Variable | Total population (n=265 300) | Patient deceased (n=30 173 11.37%) | Patient alive (n=235 127 88.63%) | OR (95% CI) | aOR† (95% CI) |
|---|---|---|---|---|---|
| **Sociodemographic variables** | | | | | |
| Age: mean (SD) | 43.40 (22.69) | 77.32 (19.06) | 39.05 (19.17) | 1.090 (1.089 to 1.091)** | 1.092 (1.090 to 1.093)** |
| Male sex: n (%) | 132 730 (50.04) | 14 848 (49.21) | 117 882 (50.15) | 0.96 (0.94 to 0.99)* | 1.48 (1.43 to 1.53)** |
| Ethnicity coded: n (%) | | | | | |
| British, Irish or any other white ethnic groups | 138 495 (58.54) | 21 993 (78.19) | 116 502 (55.89) | (reference) | (reference) |
| Mixed | 6853 (2.90) | 208 (0.74) | 6645 (3.19) | 0.17 (0.14 to 0.19)** | 0.73 (0.61 to 0.86)** |
| Indian, Pakistani, Bangladeshi or 'other Asian' | 10 889 (4.60) | 1006 (3.58) | 9883 (4.74) | 0.54 (0.50 to 0.58)** | 0.70 (0.65 to 0.76)** |
| Caribbean, African or any 'other black' | 40 725 (17.21) | 2914 (10.36) | 37 811 (18.14) | 0.41 (0.39 to 0.43)** | 0.63 (0.60 to 0.66)** |
| Other | 16 650 (7.04) | 705 (2.51) | 15 945 (7.65) | 0.23 (0.22 to 0.25)** | 0.52 (0.48 to 0.57)** |
| Not stated | 22 961 (9.71) | 1301 (4.63) | 21 660 (9.21) | 0.32 (0.30 to 0.34)** | 0.91 (0.84 to 0.98)* |
| Resident in SLaM catchment area: n (%) | 187 773 (73.42) | 23 870 (81.41) | 163 903 (72.38) | 1.67 (1.62 to 1.72)** | 0.84 (0.80 to 0.88)** |
| Quartiles of neighbourhood deprivation: n (%) | | | | | |
| First (most deprived) | 63 476 (25.03) | 7626 (26.11) | 55 850 (24.89) | (reference) | (reference) |
| Second | 63 452 (25.02) | 7049 (24.13) | 56 403 (25.14) | 0.92 (0.88 to 0.95)** | 1.08 (1.03 to 1.13)* |
| Third | 63 449 (25.02) | 7375 (25.25) | 56 074 (24.99) | 0.96 (0.93 to 1.00)* | 1.25 (1.19 to 1.31)** |
| Fourth (least deprived) | 63 221 (24.93) | 7161 (24.51) | 56 060 (24.98) | 0.94 (0.90 to 0.97)** | 1.31 (1.25 to 1.37)** |
| **Clinical variables** | | | | | |
| Primary diagnosis ever: n (%) | | | | | |
| F00-F09: organic, including symptomatic, mental disorders | 25 249 (9.52) | 12 641 (41.90) | 12 608 (5.36) | 12.73 (12.36 to 13.10)** | 1.10 (1.05 to 1.15)** |
| F10-F19: mental and behavioural disorders due to psychoactive substance use | 27 755 (10.46) | 2823 (9.36) | 24 932 (10.60) | 0.87 (0.84 to 0.91)** | 1.39 (1.31 to 1.48)** |
| F20-F29: schizophrenia, schizotypal and delusional disorders | 18 516 (6.98) | 2300 (7.62) | 16 216 (6.90) | 1.11 (1.06 to 1.17)** | 1.08 (1.01 to 1.15)* |
| F30-F39: mood (affective) disorders | 42 578 (16.05) | 5027 (16.66) | 37 551 (15.97) | 1.05 (1.02 to 1.09)* | 0.93 (0.88 to 0.97)* |
| F40-F49: neurotic, stress-related and somatoform disorders | 33 248 (12.53) | 2331 (7.73) | 30 917 (13.15) | 0.55 (0.53 to 0.58)** | 0.73 (0.69 to 0.78)** |
| F50-F59: behavioural syndromes associated with physiological disturbances and physical factors | 8743 (3.30) | 148 (0.49) | 8595 (3.66) | 0.13 (0.11 to 0.15)** | 0.47 (0.39 to 0.57)** |
| F60-F69: disorders of adult personality and behaviour | 6945 (2.62) | 415 (1.38) | 6530 (2.78) | 0.49 (0.44 to 0.54)** | 1.02 (0.91 to 1.15) |
| F70-F79: mental retardation | 3018 (1.14) | 311 (1.03) | 2707 (1.15) | 0.89 (0.79 to 1.01) | 1.24 (1.08 to 1.44)* |

Continued

**Table 4** Continued

| Variable | Total population (n=265 300) | Patient deceased (n=30 173 11.37%) | Patient alive (n=235 127 88.63%) | OR (95% CI) | aOR† (95% CI) |
|---|---|---|---|---|---|
| F80-F89: disorders of psychological development | 7242 (2.73) | 54 (0.18) | 7188 (3.06) | 0.06 (0.04 to 0.07)** | 0.65 (0.49 to 0.87)* |
| F90-F98: behavioural and emotional disorders with onset usually occurring in childhood and adolescence | 17 011 (6.41) | 77 (0.26) | 16 934 (7.20) | 0.03 (0.03 to 0.04)** | 0.45 (0.35 to 0.58)** |
| Other diagnosis | 119 638 (45.10) | 11 014 (36.50) | 108 624 (46.20) | 0.67 (0.65 to 0.69)** | 0.65 (0.63 to 0.67)** |
| Quartiles of face-to-face contacts: n (%) | | | | | |
| First (least face-to-face contact) | 76 602 (28.87) | 5926 (19.64) | 70 676 (30.06) | (reference) | (reference) |
| Second | 64 914 (24.47) | 7767 (25.74) | 57 147 (24.30) | 1.62 (1.56 to 1.68)** | 1.10 (1.04 to 1.16)** |
| Third | 59 337 (22.37) | 8798 (29.16) | 50 539 (21.49) | 2.08 (2.00 to 2.15)** | 1.11 (1.05 to 1.17)** |
| Fourth (most face-to-face contact) | 64 447 (24.29) | 7682 (25.46) | 56 765 (24.14) | 1.61 (1.56 to 1.67)** | 1.00 (0.95 to 1.06) |
| Inpatient bed days: n (%) | | | | | |
| None (0) | 238 573 (89.93) | 26 688 (88.45) | 211 885 (90.12) | (reference) | (reference) |
| Low (1–2 days) | 1446 (0.55) | 102 (0.34) | 1344 (0.57) | 0.60 (0.49 to 0.74)** | 1.52 (1.20 to 1.93)** |
| Moderate (3–31 days) | 9825 (3.70) | 1034 (3.43) | 8791 (3.74) | 0.93 (0.87 to 1.00)* | 1.58 (1.45 to 1.71)** |
| High (32+ days) | 15 456 (5.83) | 2349 (7.79) | 13 107 (5.57) | 1.42 (1.36 to 1.49)** | 1.59 (1.49 to 1.70)** |

Missing data: age (n=275); sex (n=55); ethnicity (n=28 727); resident in local catchment area (n=9537); quartiles of neighbourhood deprivation (n=11 702); referral status (n=989).
*P<0.05. **P<0.001.
†Adjusted for all other variables listed in the table.
aOR, adjusted OR; CRIS, Clinical Record Interactive Search; NHS, National Health Service; OR, Odds Ratio; SLaM, South London and Maudsley NHS Foundation Trust.

**Table 5** Logistic regression analysis examining sociodemographic and clinical associations with linked HES-ONS mortality in the matched cohort

| Variable | Total population (n=248 698) | Patient deceased (n=28 161 11.32%) | Patient alive (n=2 20 537 88.68%) | OR (95% CI) | aOR† (95% CI) | aOR‡ (95% CI) |
|---|---|---|---|---|---|---|
| **Sociodemographic variables** | | | | | | |
| Age: mean (SD) | 43.50 (22.79) | 77.75 (18.75) | 39.13 (19.28) | 1.091 (1.090 to 1.092)** | 1.093 (1.092 to 1.095)** | 1.092 (1.091 to 1.094)** |
| Male sex: n (%) | 123 818 (49.79) | 13 824 (49.09) | 109 994 (49.88) | 0.97 (0.95 to 0.99)* | 1.51 (1.45 to 1.56)** | 1.50 (1.45 to 1.55)** |
| Ethnicity coded: n (%) | | | | | | |
| British, Irish or any other white ethnic groups | 131 150 (58.93) | 20 717 (78.96) | 110 433 (56.26) | (reference) | (reference) | (reference) |
| Mixed | 6477 (2.91) | 180 (0.69) | 6297 (3.21) | 0.15 (0.13 to 0.18)** | 0.67 (0.56 to 0.80)** | 0.65 (0.54 to 0.79)** |
| Indian, Pakistani, Bangladeshi or 'other Asian' | 10 155 (4.56) | 915 (3.49) | 9240 (4.71) | 0.53 (0.49 to 0.57)** | 0.68 (0.62 to 0.74)** | 0.67 (0.62 to 0.73)** |
| Caribbean, African or any 'other black' | 38 013 (17.08) | 2611 (9.95) | 35 402 (18.03) | 0.39 (0.38 to 0.41)** | 0.60 (0.57 to 0.64)** | 0.60 (0.57 to 0.63)** |
| Other | 15 393 (6.92) | 626 (2.39) | 14 767 (7.52) | 0.23 (0.21 to 0.25)** | 0.50 (0.45 to 0.55)** | 0.50 (0.46 to 0.55)** |
| Not stated | 21 354 (9.60) | 1189 (4.53) | 20 165 (10.27) | 0.31 (0.30 to 0.33)** | 0.94 (0.87 to 1.02) | 0.92 (0.86 to 1.00)* |
| Resident in SLaM catchment area: n (%) | 177 766 (73.53) | 22 329 (81.48) | 155 437 (72.51) | 1.67 (1.62 to 1.72)** | 0.84 (0.80 to 0.88)** | 0.84 (0.80 to 0.88)** |
| Quartiles of neighbourhood deprivation: n (%) | | | | | | |
| First (most deprived) | 60 220 (25.09) | 7217 (26.42) | 53 003 (24.92) | (reference) | (reference) | (reference) |
| Second | 59 786 (24.91) | 6573 (24.06) | 53 213 (25.02) | 0.91 (0.88 to 0.94)** | 1.08 (1.03 to 1.13)* | 1.08 (1.03 to 1.13)* |
| Third | 60 052 (25.02) | 6857 (25.10) | 53 195 (25.01) | 0.95 (0.91 to 0.98)* | 1.23 (1.17 to 1.30)** | 1.23 (1.17 to 1.29)** |
| Fourth (least deprived) | 59 973 (24.99) | 6667 (24.41) | 53 306 (25.06) | 0.92 (0.89 to 0.95)** | 1.30 (1.23 to 1.36)** | 1.29 (1.23 to 1.36)** |
| **Clinical variables** | | | | | | |
| Primary diagnosis ever: n (%) | | | | | | |
| F00–F09: organic, including symptomatic, mental disorders | 24 248 (9.75) | 11 940 (42.40) | 12 308 (5.58) | 12.45 (12.09 to 12.83)** | 1.05 (1.00 to 1.11)* | 1.06 (1.01 to 1.11)* |
| F10–F19: mental and behavioural disorders due to psychoactive substance use | 26 158 (10.52) | 2549 (9.05) | 23 609 (10.71) | 0.83 (0.80 to 0.87)** | 1.35 (1.27 to 1.44)** | 1.36 (1.28 to 1.45)** |
| F20–F29: schizophrenia, schizotypal and delusional disorders | 17 323 (6.97) | 2081 (7.39) | 15 242 (6.91) | 1.07 (1.02 to 1.13)* | 1.06 (0.99 to 1.14) | 1.05 (0.98 to 1.13) |
| F30–F39: mood (affective) disorders | 40 515 (16.29) | 4715 (16.74) | 35 800 (16.23) | 1.04 (1.00 to 1.07)* | 0.92 (0.88 to 0.97)** | 0.92 (0.88 to 0.97)* |
| F40–F49: neurotic, stress-related and somatoform disorders | 31 599 (12.71) | 2187 (7.77) | 29 412 (13.34) | 0.55 (0.52 to 0.57)** | 0.73 (0.68 to 0.77)** | 0.72 (0.68 to 0.77)** |
| F50–F59: behavioural syndromes associated with physiological disturbances and physical factors | 8388 (3.37) | 136 (0.48) | 8252 (3.74) | 0.12 (0.11 to 0.15)** | 0.47 (0.38 to 0.57)** | 0.48 (0.39 to 0.58)** |
| F60–F69: disorders of adult personality and behaviour | 6585 (2.65) | 368 (1.31) | 6217 (2.82) | 0.46 (0.41 to 0.51)** | 0.99 (0.88 to 1.13) | 0.98 (0.86 to 1.12) |

Continued

**Table 5** Continued

| Variable | Total population (n=248 698) | Patient deceased (n=28 161 11.32%) | Patient alive (n=2 20 537 88.68%) | OR (95% CI) | aOR† (95% CI) | aOR‡ (95% CI) |
|---|---|---|---|---|---|---|
| F70–F79: mental retardation | 2898 (1.17) | 282 (1.00) | 2616 (1.19) | 0.84 (0.74 to 0.95)* | 1.17 (1.01 to 1.37)* | 1.17 (1.01 to 1.37)* |
| F80–F89: disorders of psychological development | 6991 (2.81) | 47 (0.17) | 6944 (3.15) | 0.05 (0.04 to 0.07)** | 0.60 (0.44 to 0.82)* | 0.61 (0.45 to 0.83)* |
| F90–F98: behavioural and emotional disorders with onset usually occurring in childhood and adolescence | 16 363 (6.58) | 67 (0.24) | 16 296 (7.39) | 0.03 (0.02 to 0.04)** | 0.42 (0.32 to 0.55)** | 0.43 (0.33 to 0.56)** |
| Other diagnosis | 112 556 (45.26) | 10 275 (36.49) | 102 281 (46.38) | 0.66 (0.65 to 0.68)** | 0.64 (0.62 to 0.66)** | 0.64 (0.62 to 0.66)** |
| Quartiles of face-to-face contacts: n (%) | | | | | | |
| First (least face-to-face contact) | 70 822 (28.48) | 5496 (19.52) | 65 326 (29.62) | (reference) | (reference) | (reference) |
| Second | 59 811 (24.05) | 7263 (25.79) | 52 548 (23.83) | 1.64 (1.58 to 1.70)** | 1.09 (1.03 to 1.15)* | 1.09 (1.04 to 1.15)* |
| Third | 56 116 (22.56) | 8282 (29.41) | 47 834 (21.69) | 2.06 (1.99 to 2.13)** | 1.11 (1.05 to 1.17)** | 1.11 (1.05 to 1.18)** |
| Fourth (most face-to-face contact) | 61 949 (24.91) | 7120 (25.28) | 54 829 (24.86) | 1.54 (1.49 to 1.60)** | 0.99 (0.93 to 1.05) | 1.00 (0.94 to 1.06) |
| Inpatient bed days: n (%) | | | | | | |
| None (0) | 223 527 (89.88) | 24 994 (88.75) | 198 533 (90.02) | (reference) | (reference) | (reference) |
| Low (1–2 days) | 1342 (0.54) | 86 (0.31) | 1256 (0.57) | 0.54 (0.44 to 0.68)** | 1.40 (1.09 to 1.81)* | 1.36 (1.04 to 1.78)* |
| Moderate (3–31 days) | 9133 (3.67) | 939 (3.33) | 8194 (3.72) | 0.91 (0.85 to 0.98)* | 1.54 (1.41 to 1.68)** | 1.53 (1.40 to 1.67)** |
| High (32+ days) | 14 696 (5.91) | 2142 (7.61) | 12 554 (5.69) | 1.36 (1.29 to 1.42)** | 1.55 (1.45 to 1.66)** | 1.55 (1.44 to 1.66)** |
| Optimal match | 203 552 (84.15) | 178 731 (87.81) | 24 821 (12.19) | 0.63 (0.61 to 0.66)** | – | 0.85 (0.81 to 0.90)** |

Missing data: sex (n=37); ethnicity (n=26 156); resident in local catchment area (n=6937); quartiles of neighbourhood deprivation (n=8667); optimal match (n=6808).

*P<0.05. **P<0.001.

†Adjusted for all other variables listed in the table (except optimal match).

‡Adjusted for optimal match rank and inverse probability weighting for matching.

aOR, adjusted OR; HES-ONS, Hospital Episode Statistics-Office for National Statistics; NHS, National Health Service; OR, Odds Ratio; SLaM, South London and Maudsley NHS Foundation Trust.

mortality model produced an OR of 1.02, whereas this effect was reversed in the linked mortality model (OR 0.98); however, neither of these findings were statistically significant.

## Post hoc analysis

In step 1 of the analysis, we found that records that did not have an NHS number or date of birth were not matched. We therefore conducted a post-hoc analysis to investigate the association between sociodemographic factors (ie, age, sex, ethnicity, resident in the SLaM catchment area and deprivation) and presence of NHS number and date of birth for matching.

With regards to NHS number, 5755 (2.17%) individuals did not have an NHS number available for matching. After conducting a multivariate logistic regression analysis (see online supplementary material table 3), we found that all the sociodemographic variables examined were significantly associated with the availability of NHS number. Specifically, older age, mixed ethnicity and being resident in the SLaM catchment area were positively associated with having an NHS number in the electronic patient record (EPR). Whereas, male sex, non-white ethnicity (except 'mixed') and the second and fourth (least deprived) quartiles of deprivation were all negatively associated with having an NHS number in the EPR.

In terms of date of birth, only 290 (0.11%) records were missing a date of birth. In univariate logistic regression analysis (see online supplementary material table 4), Asian and 'Other' ethnicities were negatively associated with having a date of birth while the second and fourth (last deprived) quartiles of deprivation were positively associated with having a date of birth in the EPR. Despite this, none of the sociodemographic variables examined were significantly associated with the availability of date of birth for matching when examined in multivariate logistic regression analysis.

## DISCUSSION
### Statement of the principal findings

By evaluating NHS EHRs linked via standard deterministic procedures to HES-ONS mortality data, we found no evidence of substantial bias distorting risk factor associations with the linked outcome measure analysed (ie, mortality). Of the 265 300 SLaM records sent to NHS Digital for matching, 93.74% were successfully matched. Despite a number of significant administrative, sociodemographic and clinical differences between records that matched and those that did not, we found minimal effects on the clinical associations with mortality. Matching error did not appear to impact the strength and direction of effects between mortality and a number of sociodemographic and clinical factors using linked mortality data after adjusting for matching probability using an inverse probability weighting statistical technique. Furthermore, after comparing the fully adjusted models predicting gold standard CRIS-derived mortality in the full cohort and linked mortality in the matched group, we found that the two models were largely the same with only slight differences in some of the ORs.

### Putting the results in context

Consistent with previous research, we found a number of significant sociodemographic differences between the matched and unmatched group including gender, ethnicity and deprivation. Previous research has demonstrated how linkage error disproportionately affects marginalised people and, therefore, missed matches result in underestimating their needs.[16 19–22 26–34] In comparison with the white ethnic group (ie, British, Irish or any other white ethnic groups), we found that individuals of all other recorded ethnicities examined were significantly less likely to match. This is potentially relevant to researchers examining health outcome variations by ethnicity and those conducting studies within similarly diverse, urban populations.[4] We found that being resident in the SLaM catchment area was not associated with match status after controlling for all other examined sociodemographic and clinical factors. Generally, patients receiving care from SLaM national specialist services, as opposed to local area services, are of increased complexity and greater clinical contact that tends to improve administrative accuracy; however, this does not seem to have had an impact on match status in the current data linkage.

This is one of the first studies to examine clinical factors associated with matching quality in a mental healthcare population and we identified a number of novel findings with regards to the clinical characteristics of the matched and unmatched groups. Patients who had not had a clinical contact with SLaM in the 2 years prior to the linkage were significantly less likely to be matched, whereas, patients with high levels of clinical contact, indicated by frequency of face-to-face contact or inpatient bed days, were significantly more likely to match. Similarly, patients who had ever received a primary diagnosis within the majority F00-98 ICD-10 diagnosis codes were significantly more likely to match than patients who had never received an ICD-10 F diagnosis. Our findings provide some indication that patients with higher severity of illness and clinical service contacts proximal to the date of matching are more likely to match, possibly through these factors driving greater administrative data accuracy. Risk factors for match status may operate through different mechanisms, for example: (1) through the type of condition and severity, (2) as determinants of linkage error or (3) both, and further research is required to understand the underlying mechanisms.

Improving the quality of data linkages, by reducing missing information in the source data, is important for scientific as well as legal and ethical reasons.[25] Within the current study, we found that records that did not contain an NHS number or date of birth in the source SLaM data were not matched. In post hoc analysis, we found that the presence of NHS number in the EPR was significantly

associated with a number of sociodemographic factors, including age, sex and ethnicity. These findings give some indication that there is a bias in the recording of NHS numbers. Future work should aim to examine the reasons why some records are missing important administrative data in order to reduce the number of missing NHS numbers and dates of birth that would subsequently help improve the missed match rate and quality of future data linkages.

The evaluation of linkage quality can guide decisions about appropriate study designs.[12] However, data linkages tend to be carried out by an independent body, and information about linkage processes are not always made available to researchers, known as the 'separation principle'.[50] By taking account of matching rank, in our sample, we found that variation in linkage quality appeared to have very little effect on the prediction of mortality. We were able to demonstrate this as the information on missed matches, and match rank was provided to us by NHS Digital. Key to health researchers being able to account for the impact of matching on the analysis of interest (in our case clinical risk factor for death) are organisations that conduct data linkages being able to provide information on match accuracy (ie, match rates, match ranks and so on) so that potential linkage error can be accounted for in subsequent analyses. In the current study, matching was on a one-to-one basis (ie, an individual was matched to a single record on the PDS); however, it is worth noting that in circumstances where matching is not one to one (eg, matching individuals to multiple episodes of care), controlling for linkage error is more problematic even when information on match accuracy is provided.

### Strengths and weaknesses of the study

This is a large-scale study of linkage error in a population of people receiving mental healthcare and one of the first studies to examine the association between patient characteristics and an optimal match. An optimal match (ie, a match rank of one – see table 2) indicates a higher quality match. Improving the number of records that achieve an optimal match is one way of improving the validity of the data linkage. By identifying factors associated with an optimal match, we can target which specific records require additional administrative work (eg, adding in missing information) to maximise the quality of the data linkage. Furthermore, the findings of this study may be generalisable to other NHS cohorts, as the quality and type of administrative data available for matching is likely to be similar within most NHS Trusts. Furthermore, standard NHS Digital matching methodology was used for the data linkage.

Gold standard data are rarely available but provide an easily interpretable way to measure linkage error[12]; a major strength of the study is the use of gold standard death data (ie, CRIS-derived mortality) as a reference category for examining error in the linked records. Within the gold standard data, we would expect the majority of deaths to be picked up either via automatic

tracing or through the Trust being informed directly, for example, via family members. However, there may be situations where the Trust is not aware of the death of a patient, for example, if the patient died outside of the UK. Furthermore, matching error due to false or missed matches is not the only reason that we may see discrepancies between the CRIS-derived mortality data and the linked mortality data from NHS Digital. For example, if the death was not registered in the UK, it would not be included in the mortality data obtained from the ONS. Similarly, there are sometimes delays in death registration due to coroner investigations that can take several months. Although SLaM may be informed at the time of death, and therefore the EPR updated to reflect this, it could be several months before the official registration takes places. Despite this, it is not always possible to identify potential missing death data or the reasons for discrepancies within the data and therefore researchers should be aware of these issues when analysing linked data.

There are some limitations of this study. Within the unmatched group, we were unable to tell apart missed matches and patients who had opted out of their data being used for secondary purposes. This is due to the strict governance requirements around opt-outs.[47] However, any researcher using data from NHS Digital will have to account for missed matching in the same way, irrespective of whether non-matching was due to opt-out. Within the current study, it was not possible for us to tease apart the impact of providing a full or part postcode on match status. In future, it may be beneficial to separate the postcode variable in order to determine whether submitting full or part postcodes has an impact on matching. Furthermore, we only examined missed matches and not false matches, and therefore, we may have underestimated the effect of linkage error. Future research should aim to examine the rate of false matches in order to gain a clearer picture of the total effect of linkage error in this population.

The data linkage in the current study was conducted by NHS Digital using deterministic matching methods; however, previous research has found that following probabilistic linkage methods can reduce the number of missed matches. With regards to HES data specifically, a previous study found that the inclusion of a probabilistic matching step to the algorithm used to link together episodes of care reduced the number of missed matches, which led to less biased estimates of hospital readmissions rates for certain patient groups such as ethnic minorities.[26] Similarly, in a data linkage between the National Neonatal Research Database and national laboratory infection surveillance data, deterministic match rates improved in later years due to patient identifiers being more complete, but despite this, probabilistic linkage was still able to identify matches not found using deterministic linkage alone.[51] Probabilistic matching is not currently a service provided as standard by NHS Digital; however going forward, the inclusion of probabilistic

matching techniques may help to reduce the number of missed matches in future refreshes of the linkage in the current study.

## Conclusion

There are significant administrative, sociodemographic and clinical differences between the matched and unmatched records in the data linkage examined. Despite this, after adjusting for matching probability, missed matches do not appear to have had an effect on the prediction of mortality. Furthermore, after comparing models predicting gold standard CRIS-derived mortality and linked mortality, we found that the models were largely the same with only slight differences in some of the ORs. Together these provide some indication that this effect is not driven by selection bias from matching error and provide some reassurance to studies using NHS linked data to investigate mortality.

**Author affiliations**
¹South London and Maudsley NHS Foundation Trust, London, UK
²Institute of Psychiatry, Psychology and Neuroscience, Kings College London, London, UK
³Centre for Paediatric Epidemiology and Biostatistics, UCL Institute of Child Health, London, UK

**Acknowledgements** We would like to acknowledge National Health Service (NHS) Digital as the provider of the HES and mortality data; those who carried out the original collection and analysis of the data bear no responsibility for their further analysis or interpretation. We would also like to acknowledge the staff at the Maudsley Biomedical Research Centre (BRC) Nucleus for their continued support of the study.

**Contributors** AJ and JD conceived the study design. Data extraction was completed by AJ with support from JD and MB. Data analysis was conducted by AJ and JD. Reporting of findings was led by AJ and JD with support from MB, RG, RDH and RS. All authors contributed to the manuscript preparation and approved the final version.

**Funding** This paper represents independent research part funded by the National Institute for Health Research (NIHR) Biomedical Research Centre (BRC) at the South London and Maudsley NHS Foundation Trust and King's College London. The work described here was part-supported by the Mental Health Data Pathfinder award to King's College London from the Medical Research Council (reference MC_PC_17214). AJ, MB, RDH and RS receive salary support from the NIHR Mental Health Biomedical Research Centre at South London and Maudsley NHS Foundation Trust and King's College London.

**Disclaimer** The views expressed are those of the authors and not necessarily those of the NHS, the NIHR, the MRC or the Department of Health.

**Competing interests** RDH has received research funding from Roche, Pfizer, Janssen and Lundbeck. The other authors have declared that no competing interests exist.

**Patient consent for publication** Not required.

**Ethics approval** Clinical Record Interactive Search (CRIS) was approved as a dataset for secondary analysis by the Oxford Research Ethics Committee C (18/ SC/0372). Approval to conduct the data linkage was granted by the Health Research Authority Confidentiality Advisory Group in July 2011 (ECC 3-04(f)/2011) and by NHS Digital's Data Access Advisory Group in October 2016 (NIC-292279-Z2S5T). Approval to conduct the study was granted in July 2017 by the CRIS Oversight Committee (17-065).

**Provenance and peer review** Not commissioned; externally peer reviewed.

**Data availability statement** Data may be obtained from a third party and are not publicly available. Data are owned by a third party, Maudsley Biomedical Research Centre (BRC) Clinical Records Interactive Search (CRIS) tool, which provides access to anonymised data derived from SLaM electronic medical records. These data can only be accessed by permitted individuals from within a secure firewall (ie,

the data cannot be sent elsewhere), in the same manner as the authors. For more information please contact: cris.administrator@slam.nhs.uk.

**ORCID iDs**
Amelia Jewell http://orcid.org/0000-0002-0887-2159
Johnny Downs http://orcid.org/0000-0002-8061-295X

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
