## [Reviewer comments · BMJ Open]

ARTICLE DETAILS

TITLE (PROVISIONAL)	The Impact of Matching Error on Linked Mortality Outcome in a Data Linkage of Secondary Mental Health Data with Hospital Episode Statistics (HES) and Mortality Records in South East London: A Cross-Sectional Study
AUTHORS	Jewell, Amelia; Broadbent, Matthew; Hayes, Richard; Gilbert, Ruth; Stewart, Robert; Downs, Johnny

VERSION 1 - REVIEW

REVIEWER	Marcos Ennes Barreto Federal University of Bahia (UFBA), Salvador, Brazil
REVIEW RETURNED	08-Dec-2019

GENERAL COMMENTS	This paper intends to measure the impact of matching errors during administrative (routine) data linkage using a case study comprising mental health x mortality records. The authors conclude that estimates related to mortality were minimal between matched and unmatched records, despite other significant differences related to social and clinical determinants. The authors also claim novelty and contributions related to i) the use of gold standard mental health records linked to specific outcomes (ex. death) and ii) generalizability of their approach among other NHS settings (although restricted to the UK). - Pg 5, line 41: recording of deaths are informed by family members or tracked back through SCR. Can we assume that missing notifications (from family members) will be fully recovered by the SCR tracking system or there is a significant number of missing values (related to death registration)? - The analysis related to diagnosis has potentially found exclusion criteria (no association with matching status) associated with codes F20-F29 and F60-F69 that worth further investigation. All other analyses are presented in terms of more or less propensity to match, which means some level of association. Table 1: - postcode present (part or full) => what is the impact on linkage accuracy between part and full situations?- What is the impact of the catchment area in the analysis? Is it to indicate imported deaths?- Primary diagnosis ever: if a patient has multiple episodes within SLaM and deceased, his death will be associated with the last diagnosis or it would be necessary to further investigate how
---

	multiple (sequential) conditions have affected him? Table 2: what is the policy about Postcode in rule 6? If the postcode is in the ignore list, what it means the 'exact' result? Both postcodes shall be on the ignore list? It is also not so clear when comparing to rule 7. I am curious about using machine learning (ML) models to stratify patients according to some socio-demographic, administrative and clinical attributes, as the authors have pointed out situations where some attributes do influence the results and situations where they do not influence (in a very intermittent form). It seems, in a broader view, a bit complex to devise high-accurate ML models over these linked datasets. Overall, the paper brings very interesting findings related to i) differences in socio-demographic, administrative and clinical determinants of matching/unmatching status, ii) similarity of models using gold standard and linked datasets, and iii) presence of missed matches not affecting the outcomes. All of these findings carry an important contribution to decision making and should be further evaluated against other (similar) case studies. The authors claim generalizability throughout the NHS system, so new partnerships intending to reproduce this study and compare results are highly indicated before any definitive assumption. As there is some restriction related to data sharing, it is advised to reproduce this study with similar data from other NHS Trusts. I would also suggest to further investigate those administrative, socio-demographic and clinical variables potentially not affecting the outcomes to check for the reproducibility of findings. Typos: - Pg 4, line 46, ...35], providing some indication... - Pg 4, lines 47 to 53: the two sentences on the usage of gold standard datasets are too similar. I suggest rewriting and/or collapsing them (if possible) into one sentence. The reviewer provided a marked copy with additional comments. Please contact the publisher for full details.
--	--

REVIEWER	Sean Randall Curtin University
REVIEW RETURNED	18-Dec-2019

GENERAL COMMENTS	This study examined the effects of errors of linkage quality (missed matches only) - how they effect study results in one particular example, and the socio-demographic groups which are most effective. The paper is well written and can be published as is. The main concern is perhaps a lack of novelty - however novelty is not a criteria for this journal so this paper is appropriate. Major comments: There appears almost no analysis as to the greater issue of why certain sociodemographic groups have reduced data quality. Do people not have/know their NHS numbers? Do people not know their
---

	date of birth? Is it simply due to these details existing but not being recorded? If so, do certain locations have poorer recording rates than others (maybe those in poorer areas?) Location of service seems the most obvious predictor of poor data quality that is not included in this study. Further analysis along exploring the reasons why data quality may be worse for certain groups is needed. Minor comments:  - Introduction, third paragraph 'Providing some indication' stem - consider revising sentence/joining with previous sentence. - The last sentence of the results should be removed - with odds ratios of 1.02 and 0.98, it is not appropriate to talk of less or more likely to have died. - Last sentence of 'results in context'. May be worth noting that in many (most?) circumstances (when matching is not 1:1), we do not know how to control for linkage error, even if information on match accuracy is provided.
--	--

VERSION 1 – AUTHOR RESPONSE

Reviewer 1:

1. Pg 5, line 41: recording of deaths are informed by family members or tracked back through SCR. Can we assume that missing notifications (from family members) will be fully recovered by the SCR tracking system or there is a significant number of missing values (related to death registration)?

RESPONSE: Reviewer 1 raises an important point, we would expect the majority of cases to be fully recovered by the SCR tracking, however, there may be situations where deaths could be missed, for example, if an individual died outside of the UK. Although, it is very difficult for us to know how many records we may be missing. We have added some additional detail to the discussion regarding this (Page 13):

“Within the gold-standard data we would expect the majority of deaths to be picked up either via automatic tracing or through the Trust being informed directly (for example via family members), however there may be situations where the Trust is not aware of the death of a patient, for example if the patient died outside of the UK. Furthermore, matching error due to false or missed matches is not the only reason that we may see discrepancies between the CRIS-derived mortality data and the linked mortality data from NHS Digital. For example, if the death was not registered in the UK it would not be included in the mortality data obtained from the ONS. Similarly, there are sometimes delays in death registration due to coroner investigations which can take several months. Although SLaM may be informed at the time of death, and therefore the EPR updated to reflect this, it could be several months before the official registration takes places. Despite this, it is not always possible to identify potential missing death data or the reasons for discrepancies within the data and therefore researchers should be aware of these issues when analysing linked data.”

2. The analysis related to diagnosis has potentially found exclusion criteria (no association with matching status) associated with codes F20-F29 and F60-F69 that worth further investigation. All other analyses are presented in terms of more or less propensity to match, which means some level of association.

RESPONSE: We found diagnostic codes F20-F29 and F60-F69 were not significantly associated with match status. All we can conclude from these findings is that within a clinical sample these particular diagnoses do no confer an increased or decreased likelihood for matching after adjustment for other

potential confounders, including other prior and/or coexisting diagnoses. Given the size of the sample, the likelihood of a Type 2 error is minimal.

3. Table 1:

a. Postcode present (part or full) => what is the impact on linkage accuracy between part and full situations?

RESPONSE: We agree with the reviewer that it would be interesting to explore the impact of part and full postcodes on match status. Unfortunately due to the governance around the use of Patient Identifiable Information (PII) we no longer have access to the original identifiers sent to NHS Digital for linkage. We are unable to differentiate between records with full or part postcodes, we only have a flag which indicates whether any postcode (full or part) was available for matching. We agree that this would be an interesting factor to consider for future projects and have added some additional detail to the Discussion regarding this (Page 14):

“Within the current study, it was not possible for us to tease apart the impact of providing a full or part postcode on match status. In future it may be beneficial to separate the postcode variable in order to determine the impact of full and part postcodes on matching.”

b. What is the impact of the catchment area in the analysis? Is it to indicate imported deaths?

RESPONSE: SLaM provides local area services for the London Boroughs of Lambeth, Southwark, Lewisham and Croydon (the SLaM ‘catchment area’) as well as some national specialist services. Generally those receiving national level care are of increased severity and complexity which may have a bearing both on the likelihood of accurate administrative record keeping and likelihood of death. We have made this clearer in the manuscript by adding additional detail to the Methods section (Page 5):

“SLaM provides local area mental health services predominately for the London boroughs of Lambeth, Southwark, Croydon, and Lewisham (the SLaM ‘catchment area’), as well as some national specialist services”

We have also updated the Discussion (Page 12):

“We found that being resident in the SLaM catchment area was not associated with match status after controlling for all other examined sociodemographic and clinical factors. Generally, patients receiving care from SLaM national specialist services, as opposed to local area services, are of increased complexity and greater clinical contact which tends to improve administrative accuracy however, this does not seem to have had an impact on match status in the current data linkage.”

c. Primary diagnosis ever: if a patient has multiple episodes within SLaM and deceased, his death will be associated with the last diagnosis or it would be necessary to further investigate how multiple (sequential) conditions have affected him?

RESPONSE: Within the current study we looked at all primary diagnoses received by individuals, which in some cases would include multiple diagnoses per person. We agree with the Reviewer that the association between multiple diagnoses and death is an important area of study, however, we felt that it was outside of the scope of the current study as the focus was more on the impact of linkage error on the prediction of mortality rather than the prediction of mortality itself.

4. Table 2: what is the policy about Postcode in rule 6? If the postcode is in the ignore list, what it means the ‘exact’ result? Both postcodes shall be on the ignore list? It is also not so clear when comparing to rule 7.

RESPONSE: We are grateful to the Reviewer for bringing this to our attention, the 'ignore' list includes postcodes for communal establishments such as hospitals, care homes, prisons, military bases, student halls of residence, and boarding schools. For both match rank 6 and 7 the postcode must be an exact match, the difference between the two match ranks is that those individuals who match in rank 6 have postcodes which are associated with communal establishments. We have updated the foot note of the table to clarify this (Page 26, Table 2):

"The 'ignore' list includes postcodes for communal establishments such as hospitals, care homes, prisons, and boarding schools."

5. I am curious about using machine learning (ML) models to stratify patients according to some socio-demographic, administrative and clinical attributes, as the authors have pointed out situations where some attributes do influence the results and situations where they do not influence (in a very intermittent form). It seems, in a broader view, a bit complex to devise high-accurate ML models over these linked datasets.

RESPONSE: We would like to thank the Reviewer for their interesting comment. Unfortunately it was not within the scope of the current study to consider the use of Machine Learning models to stratify patients according to their socio-demographic, administrative, and clinical attributes, however, if the reviewer would like to contact us separately we would be pleased to explore the possible applications of machine learning on our data.

6. I would also suggest to further investigate those administrative, socio-demographic and clinical variables potentially not affecting the outcomes to check for the reproducibility of findings.

RESPONSE: We agree with the reviewer that our findings are important to replicate. We conducted a large scale study on a whole population dataset of people receiving mental healthcare at the South London and Maudsley NHS Foundation Trust which means that our findings may be generalisable to other NHS cohorts to an extent, however, we welcome any further replication work using linked administrative data in different settings and we have included this in our discussion (Page 14):

"The study sample was largely derived from a diverse urban and semi-urban population in South London, and is therefore not nationally representative. However, the findings may be generalisable to other NHS cohorts, as the quality and type of administrative data available for matching is likely to be similar within most NHS Trusts, and national standard NHS Digital matching methodology was used for the data linkage. Future studies should aim to replicate this work using linked administrative data in different settings in order to check for the reproducibility of findings."

7. Typos:

- a. Pg 4, line 46, ...35], providing some indication...
- b. Pg 4, lines 47 to 53: the two sentences on the usage of gold standard datasets are too similar. I suggest rewriting and/or collapsing them (if possible) into one sentence.

RESPONSE: We are grateful to the reviewer for drawing these points to our attention; these have now been amended in the manuscript.

Reviewer 2:

1. There appears almost no analysis as to the greater issue of why certain sociodemographic groups have reduced data quality. Do people not have/know their NHS numbers? Do people not know their date of birth? Is it simply due to these details existing but not being recorded? If so, do certain locations have poorer recording rates than others (maybe those in poorer areas?) Location of service

seems the most obvious predictor of poor data quality that is not included in this study. Further analysis along exploring the reasons why data quality may be worse for certain groups is needed.

RESPONSE: Reviewer 2 raises an important point; the issues of why certain individuals have reduced administrative data quality is important to understand in order to improve data quality and subsequently linkage match rates. In Step 1 of the analysis we found that records which did not have an NHS number or date of birth were not matched. In light of the Reviewers feedback we conducted a post-hoc analysis to investigate the association between sociodemographic factors (i.e. age, sex, ethnicity, resident in the SLaM catchment area, and deprivation) and presence of NHS number and date of birth for matching.

With regards to NHS number, 5755 (2.17%) individuals did not have an NHS number available for matching. After conducting a multivariate logistic regression analysis, we found that all the sociodemographic variables examined were significantly associated with the availability of NHS number. Specifically, older age, mixed ethnicity, and being resident in the SLaM catchment area were positively associated with having an NHS number in the Electronic Patient Record (EPR). Whereas, male sex, non-white ethnicity (except 'mixed') and the 2nd-4th (least deprived) quartiles of deprivation were all negatively associated with having an NHS number in the EPR.

In terms of date of birth, only 290 (0.11%) records were missing a date of birth. In univariate analysis Asian and 'Other' ethnicities were negatively associated with having a date of birth whilst the 2nd and 4th (last deprived) quartiles of deprivation were positively associated with having a date of birth in the EPR. Despite this, none of the sociodemographic variables examined were significantly associated with the availability of date of birth for matching when examined in multivariate analysis.

It is important to note that it is not possible for us to tell whether individuals know their own NHS number and/or date of birth. The outcome examined relates to the presence of these variables in the patient's record and there may be many reasons why information is not recorded in an individual's record.

We have updated the Results section of the manuscript to include detail of the post-hoc analysis (Page 10) and we have included the additional analyses in the Supplementary Material (Table 2 and 3):

"In Step 1 of the analysis we found that records which did not have an NHS number or date of birth were not matched. We therefore conducted a post-hoc analysis to investigate the association between sociodemographic factors (i.e. age, sex, ethnicity, resident in the SLaM catchment area, and deprivation) and presence of NHS number and date of birth for matching.

With regards to NHS number, 5755 (2.17%) individuals did not have an NHS number available for matching. After conducting a multivariate logistic regression analysis (see Table 2 in Supplementary Material), we found that all the sociodemographic variables examined were significantly associated with the availability of NHS number. Specifically, older age, mixed ethnicity, and being resident in the SLaM catchment area were positively associated with having an NHS number in the electronic patient record (EPR). Whereas, male sex, non-white ethnicity (except 'mixed') and the 2nd and 4th (least deprived) quartiles of deprivation were all negatively associated with having an NHS number in the EPR.

In terms of date of birth, only 290 (0.11%) records were missing a date of birth. In univariate logistic regression analysis (see Table 3 in Supplementary Material) Asian and 'Other' ethnicities were negatively associated with having a date of birth whilst the 2nd and 4th (last deprived) quartiles of deprivation were positively associated with having a date of birth in the EPR. Despite this, none of the

sociodemographic variables examined were significantly associated with the availability of date of birth for matching when examined in multivariate logistic regression analysis.”

We have also added further detail to the Discussion (Page 12):

“Improving the quality of data linkages, by reducing missing information in the source data, is important for scientific as well as legal and ethical reasons [25]. Within the current study we found that records which did not contain an NHS number or date of birth in the source SLAM data were not matched. In post-hoc analysis we found that the presence of NHS number in the EPR was significantly associated with a number of sociodemographic factors, including age, sex, and ethnicity. These findings give some indication that there is a bias in the recording of NHS numbers. Future work should aim to examine the reasons why some records are missing important administrative data in order to reduce the number of missing NHS numbers and dates of birth which would subsequently help improve the missed match rate and quality of future data linkages.”

2. Introduction, third paragraph 'Providing some indication" stem - consider revising sentence/joining with previous sentence.

RESPONSE: Thank you for drawing this to our attention; we have amended this in the manuscript.

3. The last sentence of the results should be removed - with odds ratios of 1.02 and 0.98, it is not appropriate to talk of less or more likely to have died.

RESPONSE: We have removed this sentence and replaced it with the following (Page 10):

“The association between a primary diagnosis of a disorder of adult personality and behaviour (F60-F69) and mortality in the CRIS-derived mortality model produced an odds ratio of 1.02, whereas this effect was reversed in the linked mortality model (OR: 0.98); however, neither of these findings were statistically significant.”

4 Last sentence of 'results in context'. May be worth noting that in many (most?) circumstances (when matching is not 1:1), we do not know how to control for linkage error, even if information on match accuracy is provided.

RESPONSE: We agree with the reviewer that when matching is not one to one it is difficult to know how to control for linkage error. We have added the following to the discussion (Page 13).

“In the current study, matching was on a one to one basis (i.e. an individual was matched to a single record on the PDS), however, it is worth noting that in circumstances where matching is not one to one (e.g. matching individuals to multiple episodes of care) controlling for linkage error is more problematic even when information on match accuracy is provided.”

VERSION 2 – REVIEW

REVIEWER	Marcos Ennes Barreto Federal University of Bahia (UFBA) Salvador, BA, Brazil
REVIEW RETURNED	02-Mar-2020

GENERAL COMMENTS	This revised version has improved substantially and the authors have presented satisfactory answers and adjustments to all my
---

	previous suggestions and comments. I have no additional suggestions for this paper. The authors still point some limitations in their study related to i) regional settings (South East London) x representativeness and ii) missing data due to opt-out. I wouldn't put them on the list as they are a bit out of reach of the authors. Any sample (or setting) won't be sufficiently representative. Records with partial data (due to opt-out) will always exist and should be inserted/discarded from the study following some criteria. This discussion on limitations should focus, in my opinion, on using deterministic x probabilistic approaches. Ideally, this study should be compared with similar ones from different settings/regions or using other databases with similar data.
--	--

REVIEWER	Sean Randall Curtin University
REVIEW RETURNED	19-Feb-2020

GENERAL COMMENTS	Thank you for addressing my queries
-------------------------------------

VERSION 2 – AUTHOR RESPONSE

Reviewer 1:

1. The authors still point some limitations in their study related to i) regional settings (South East London) x representativeness and ii) missing data due to opt-out. I wouldn't put them on the list as they are a bit out of reach of the authors. Any sample (or setting) won't be sufficiently representative. Records with partial data (due to opt-out) will always exist and should be inserted/discarded from the study following some criteria. This discussion on limitations should focus, in my opinion, on using deterministic x probabilistic approaches. Ideally, this study should be compared with similar ones from different settings/regions or using other databases with similar data.

RESPONSE: Reviewer 1 raises an important point; probabilistic matching techniques have previously been found to reduce the number of missed matches and therefore may provide a method for reducing linkage bias in large database linkages such as this. In light of the reviewer's suggestions we have amended the limitations section of the discussion and added in further detail around probabilistic matching. We have also removed the limitation of representativeness; although, we have retained the detail around opt-out as we feel it is important for individuals who use the linked data to be aware of this (page 13):

“This is a large-scale study of linkage error in a population of people receiving mental healthcare and one of the first studies to examine the association between patient characteristics and an optimal match. An optimal match (i.e. a match rank of one - see Table 2) indicates a higher quality match. Improving the number of records which achieve an optimal match is one way of improving the validity of the data linkage. By identifying factors associated with an optimal match we can target which specific records require additional administrative work (e.g. adding in missing information) to maximise the quality of the data linkage. Furthermore, the findings of this study may be generalisable to other NHS cohorts, as the quality and type of administrative data available for matching is likely to be similar within most NHS Trusts. Furthermore, standard NHS Digital matching methodology was used for the data linkage.

Gold standard data are rarely available but provide an easily interpretable way to measure linkage error [12], a major strength of the study is the use of gold standard death data (i.e. CRIS-derived mortality) as a reference category for examining error in the linked records. Within the gold-standard data we would expect the majority of deaths to be picked up either via automatic tracing or through the Trust being informed directly, for example via family members. However, there may be situations where the Trust is not aware of the death of a patient, for example if the patient died outside of the UK. Furthermore, matching error due to false or missed matches is not the only reason that we may see discrepancies between the CRIS-derived mortality data and the linked mortality data from NHS Digital. For example, if the death was not registered in the UK it would not be included in the mortality data obtained from the ONS. Similarly, there are sometimes delays in death registration due to coroner investigations which can take several months. Although SLAM may be informed at the time of death, and therefore the EPR updated to reflect this, it could be several months before the official registration takes place. Despite this, it is not always possible to identify potential missing death data or the reasons for discrepancies within the data and therefore researchers should be aware of these issues when analysing linked data.

There are some limitations of this study. Within the unmatched group, we were unable to tell apart missed matches and patients who had opted out of their data being used for secondary purposes. This is due to the strict governance requirements around opt-outs [47]. However, any researcher using data from NHS Digital, will have to account for missed matching in the same way, irrespective of whether non-matching was due to opt-out. Within the current study, it was not possible for us to tease apart the impact of providing a full or part postcode on match status. In future, it may be beneficial to separate the postcode variable in order to determine whether submitting full or part postcodes has an impact on matching. Furthermore, we only examined missed matches and not false matches, and therefore we may have underestimated the effect of linkage error. Future research should aim to examine the rate of false matches in order to gain a clearer picture of the total effect of linkage error in this population.

The data linkage in the current study was conducted by NHS Digital using deterministic matching methods; however, previous research has found that following probabilistic linkage methods can reduce the number of missed matches. With regards to HES data specifically, a previous study found that the inclusion of a probabilistic matching step to the algorithm used to link together episodes of care reduced the number of missed matches which led to less biased estimates of hospital readmissions rates for certain patient groups such as ethnic minorities [26]. Similarly, in a data linkage between the National Neonatal Research Database and national laboratory infection surveillance data, deterministic match rates improved in later years due to patient identifiers being more complete, but despite this, probabilistic linkage was still able to identify matches not found using deterministic linkage alone [51]. Probabilistic matching is not currently a service provided as standard by NHS Digital, however going forward, the inclusion of probabilistic matching techniques may help to reduce the number of missed matches in future refreshes of the linkage in the current study.”